# Genetic and Morphological Approach for Western Corn Rootworm Resistance Management

**Martina Kadoić Balaško** [1,*]🄳, **Katarina M. Mikac** [2]🄳, **Hugo A. Benítez** [3]🄳, **Renata Bažok** [1]🄳 **and Darija Lemic** [1]🄳

1   Department of Agricultural Zoology, Faculty of Agriculture, University of Zagreb, Svetošimunska 25, 10000 Zagreb, Croatia; rbazok@agr.hr (R.B.); dlemic@agr.hr (D.L.)
2   Centre for Sustainable Ecosystem Solutions, Faculty of Science, Medicine and Health, School of Biology, University of Wollongong, Wollongong, NSW 2522, Australia; kmikac@uow.edu.au
3   Centro de Investigación de Estudios Avanzados del Maule, Laboratorio de Ecología y Morfometría Evolutiva, Universidad Católica del Maule, Talca 3466706, Chile; hbenitez@ucm.cl
*   Correspondence: mbalasko@agr.hr; Tel.: +385-1239-3670

**Abstract:** The western corn rootworm (WCR), is one of the most serious pests of maize in the United States. In this study, we aimed to find a reliable pattern of difference related to resistance type using population genetic and geometric morphometric approaches. To perform a detailed population genetic analysis of the whole genome, we used single nucleotide polymorphisms (SNPs) markers. For the morphometric analyses, hindwings of the resistant and non-resistant WCR populations from the US were used. Genetic results showed that there were some differences among the resistant US populations. The low value of pairwise $F_{ST}$ = 0.0181 estimated suggests a lack of genetic differentiation and structuring among the putative populations genotyped. However, STRUCTURE analysis revealed three genetic clusters. Heterozygosity estimates ($H_O$ and $H_E$) over all loci and populations were very similar. There was no exact pattern, and resistance could be found throughout the whole genome. The geometric morphometric results confirmed the genetic results, with the different genetic populations showing similar wing shape. Our results also confirmed that the hindwings of WCR carry valuable genetic information. This study highlights the ability of geometric morphometrics to capture genetic patterns and provides a reliable and low-cost alternative for preliminary estimation of population structure. The combined use of SNPs and geometric morphometrics to detect resistant variants is a novel approach where morphological traits can provide additional information about underlying population genetics, and morphology can retain useful information about genetic structure. Additionally, it offers new insights into an important and ongoing area of pest management on how to prevent or delay pest evolution towards resistant populations, minimizing the negative impacts of resistance.

**Keywords:** *Diabrotica virgifera virgifera*; *Bt* toxins; resistance; geometric morphometrics; SNPs

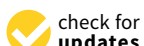



## 1. Introduction

Maize (*Zea mays* L.) is one of the most important crops worldwide. About 200 million hectares is planted, with an average yield of 22 tons/hectare, resulting in 1150 million tons of maize harvested worldwide [1]. The western corn rootworm (WCR) *Diabrotica virgifera virgifera* is the worst pest in the United States and a major alien invasive pest in Europe [2,3]. The main damage caused by WCR to maize plants is by its larval stage that feeds on corn roots, which affects important physiological processes of the plant. The resulting damage leads to stalk lodging and yield losses, which in turn leads to economic damage to crops [4].

Suppression with chemical insecticides is an important management tool for this pest [5], but WCR has rapidly developed resistance to the insecticides used for control [6]. The first noted case of resistance to insecticides was to cyclodiene insecticides (aldrin and heptachlor) in 1959 in Nebraska [7,8]. So far, WCR has evolved resistance to organophosphates (methyl parathion), carbamates (carbaryl) [6,9], and pyrethroids (bifenthrin and

tefluthrin) [10,11]. In addition to insecticides, WCR has developed resistance to crop rotation [12–14] and to the *Bt* toxin in genetically modified maize [15]. Crop rotation remains the most effective control tactic against WCR. However, resistance to crop rotation has been documented in Illinois and other neighboring states [12]. Spencer et al. [16] observed that some of the WCR populations in northern Indiana and east central Illinois feed on soya bean foliage and flowers, as well as lay eggs in soya bean fields. This behavioral change in the WCR populations in the eastern Corn Belt has eliminated the effectiveness of crop rotation as a rootworm management option. As a consequence, the use of soil and foliar insecticides for WCR has increased to protect corn following soya bean. It was estimated that each year WCR costs US farmers at least USD 1 billion through yield losses and treatment costs [17], but after adaptation to crop rotation, these losses are estimated to be higher [18]. Transgenic maize expressing *Bacillus thuringiensis* (*Bt*) was introduced in 2003 in the United States [15]. However, resistance to maize expressing Cry3Bb1 was reported in Iowa in 2009 [19]. Afterwards, resistance to Cry3Bb1 was detected in fields throughout Iowa [20,21] but also in WCR populations found in Illinois, Nebraska, and Minnesota [22–24]. Selected rootworm populations developed resistance to the toxins Cry34/Cry35Ab1, Cry3Bb1, and mCry3A under laboratory and greenhouse conditions [25–28]. Cross-resistance was found in WCR field populations between the Cry3Bb1, mCry3A, and eCry3.1Ab toxins [21–23,29]. WCR populations evolved resistance to all four currently available Bt toxins (Cry3Bb1, mCry3A, eCry3.1Ab, and Cry34/35Ab1) [19,23,29–31], and consequently, the challenge of managing has become more difficult.

Resistance is a dynamic phenomenon, meaning that mechanisms already known can change over time. Ongoing monitoring is essential to determine whether management recommendations remain valid or need to be revised in light of changing circumstances or newly acquired knowledge [32]. WCR resistance to insecticides and management strategies is a serious and growing problem in maize production, and before it becomes an even more widespread and major problem, there is a need to explore and implement novel methods (such as single nucleotide polymorphisms and geometric morphometrics) for the early detection of resistance or adaptation that causes WCR resistance.

Population genetic markers can be used to provide genetic data for WCR that is useful when investigating changes in genetic structure and differentiation [3,33,34]. Different types of molecular markers (allozymes, mtDNA sequencing, AFLPs, microsatellites, and SNPs) have already been used in North American WCR populations. The result showed high genetic diversity and a general lack of population structure across the US Corn Belt [35–37].

Several studies on WCR resistance mechanisms have been performed [38–40]. Coates et al. [41] attempted the use of SNPs as population genetic markers in WCR in the US and showed that both markers (microsatellites and SNPs) gave similar results. This does not suggest that SNPs are less effective at separating genetic variation in the species, but it is likely a result of low numbers of SNPs and low genome coverage because the authors used 12 biallelic loci among 190 individuals. Wang et al. [40] found that cylcodiene resistance is correlated with SNPs in the gamma-aminobutyric acid (GABA) receptor. Flagel et al. [42] used SNPs to identify candidate gene families for insecticide resistance and to understand how population processes have shaped variation in WCR populations. Their WCR transcriptome assembly included several gene families that have been implicated in insecticide resistance in other species and that have provided a foundation for future research. Flagel et al. [43] discovered and validated genetic markers in WCR associated with resistance to the *Bt* toxin Cry3Bb1. They found that the inheritance of Cry3Bb1 resistance is associated with a single autosomal linkage group and is almost completely recessive. Niu et al. [44] found that SNP markers identified in a single autosomal linkage group (LG8, 115–135 cm) were correlated with resistance to Cry3Bb1 in field populations of WCR. Although the linkage of these genes to Cry3Bb1 resistance was strong, the causal gene for Cry3Bb1 resistance was not confirmed and remains to be reported.

Geometric morphometrics (GM) (i.e., phenotype size and shape analysis) is a technique that can be used to show hindwing shape and size differences among rootworm populations [45]. By analyzing wing size and shape, it is possible to reveal the invasive adaptation of the adults' traits to different environmental influences. Numerous studies have been performed on the WCR hindwings using geometric morphometry [46–49]. Mikac et al. [46] provided preliminary evidence of wing shape and size differences in WCR from rotated versus continuous maize. Most recently, Mikac et al. [45] determined morphological differences in wing shape in populations adapted to crop rotation and *Bt* maize compared with a non-resistant WCR population. This study showed evidence of differential wing shape in relation to resistance development and highlights the importance of wing size and shape as a reliable, inexpensive, yet effective biomarker for resistance detection in corn rootworm. The research of Mikac et al. [45] looked at the *Bt*-resistant individuals as a whole, so it is necessary to extend their research to each *Bt* toxin separately. A deeper understanding of maize rootworm wing shape and flight morphology, wing geometry, aspect ratio, and flight efficiencies will help identify which resistant phenotypes are most likely to invade geographic areas where they are not yet present.

According to Bouyer et al. [50], changes in an organism's genotype takes much longer to manifest than in its phenotype, thus making geometric morphometrics a much more useful tool than genetics for detecting changes in populations in the short term. That suggests morphology can retain useful information on genetic structure and has the benefit over molecular methods of being inexpensive, easy to use, and able to yield a lot of information quickly. However, resistance cannot be fully understood without genetic data. Genetic studies are an important tool for developing improved methods for detecting resistance, for studying resistance mechanisms, and for choosing approaches to resistance management [51]. Several studies suggest that results are more accurate when both methods are combined. Morphological traits can provide additional information about underlying population genetics, and morphology can retain useful information about genetic structure [52–56].

This is the first study that combines both genetic and geometric morphometric techniques on the same WCR populations and same individuals. The aim of this study was to define genetic variables between known phenotypes and to explore phenotypic markers related to changes in the genome. We hypothesized that by combining genetic and morphological markers, it would be possible to determine and predict resistance to *Bt* toxins and crop rotation in the field.

## 2. Materials and Methods

### 2.1. Sample Collection

All WCR individuals used in this research were populations from the US. The same individuals were used both for the genetic and morphometric analysis. WCR individuals were collected from South Dakota in the fields containing transgenic corn. Individuals adapted to crop rotation from Illinois were collected in fields with documented resistance. Non-resistant (susceptible) adults were obtained from the NCARL laboratory. The non-resistant laboratory population was originally collected in 1987 near the town of Trent, South Dakota, in Moody County. It has been in continuous rearing since that time without mixing with other collections. It is approximately one generation per year. The original beetles were selected in cornfields or on the edge of cornfields and the adult beetles were returned to the laboratory. The non-resistant colony is reared in soil on maize roots and the adult beetles are fed on an artificial diet. Attempts are being made to keep the rearing protocol "field-like" to keep it "wild" (Chad Nielson personal communication). According to Mikac et al. [45], there are minimal differences between rotation-resistant laboratory and field-collected populations, suggesting that the rearing system was not the main reason for the differences observed in their study. Therefore, we excluded the possibility that different conditions (field, laboratory rearing) may contribute to differences in wing shapes and sizes.

Individuals were placed in 95% ethanol pending genetic and morphometric analysis. WCR individuals used in this research were adapted to crop rotation, were non-resistant, and were collected from *Bt* corn expressing different toxins (Cry3Bb1, Cry34/35Ab1, Cry3Bb1, and Cry34/35Ab1) (Table 1).

**Table 1.** Number of WCR individuals used for geometric morphometric and SNPs analyses. *n* = sample size.

| Western Corn Rootworm Populations | Geometric Morphometric Wings (*n*) | Males/ Females | Adults Single Nucleotide Polymorphisms Genotyped (*n*) | Males/ Females |
|---|---|---|---|---|
| Cry3Bb1 | 433 | 184/252 | 7 | 2/5 |
| Cry3Bb1_Cry34/35Ab1 | 86 | 27/59 | 5 | 3/2 |
| Cry34/35Ab1 | 91 | 32/59 | 6 | 3/3 |
| Adapted to crop rotation | 31 | 14/17 | 4 | 1/3 |
| Non-resistant | 134 | 66/68 | 7 | 4/3 |

*2.2. DNA Extraction and SNPs Genotyping*

Before DNA extraction, hindwings from all individuals were removed for morphometric analysis. DNA was then extracted from the whole-body tissue of 29 adult WCR. DNA extractions were performed using the Qiagen DNeasy Blood and Tissue Kit (QIAGEN, Hilden, Germany) following the manufacturer's protocol.

The DNA concentration for all samples was measured using spectrophotometer (BioSpec-nano Micro-volume) and adjusted to 50 ng/µL prior to SNPs genotyping by Diversity Arrays Technology (DArT) [57,58]. After quality control, 29 samples were sent for genotyping. Genotyping was undertaken by Diversity Array Technology Pty Ltd. (DArT, Canberra, Australia) using the extracted WCR DNA. This method is based on methyl filtration and next-generation sequencing platforms [58]. The data we received were filtered for minor allele frequency (MAF) lower than 0.1 and also for missing data higher than 10%. Quality of SNP markers was determined by the parameters "reproducibility" and "call rate" [59]. Remaining SNPs were used for further analysis of genetic diversity and population structure.

*2.3. Geometric Morphometric Sample Preparation*

The adult WCRs (see Table 1) were investigated using geometric morphometric procedures and analyses based on hindwing venation undertaken. In total, 775 hindwings of WCR were analyzed. Left and right hindwings were removed from each individual and slide-mounted using the fixing agent Euparal (Carl Roth GmbH + Co. KG, Karlsruhe, Germany) based on standard methods [60]. Slide-mounted wings were photographed using a Canon PowerShot A640 digital camera (10-megapixel) on a trinocular mount of a Zeiss Stemi 2000-C Leica stereo-microscope and saved in JPEG format using the Carl Zeiss AxioVision Rel. 4.6. (Carl Zeiss Microscopy GmbH, München, Germany). Fourteen type 1 landmarks defined by vein junctions or vein terminations were used (Figure 1.) [47–49,61].

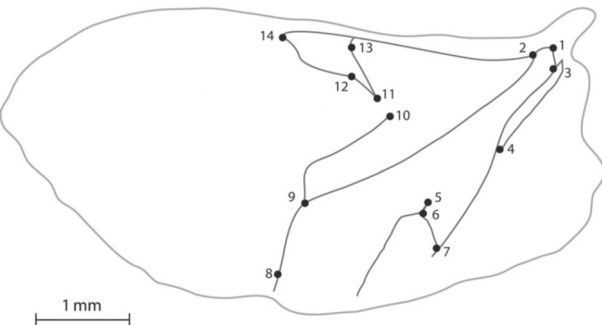

**Figure 1.** Representation of the 14 morphological landmarks identified on the hindwings of western corn rootworm [61].

*2.4. Data Analysis*

2.4.1. Genetic Data

All population genetic data analyses were undertaken using the coding environment in R using the R packages adegenet v2.1.3 [62] and dartR v1.1.11 [63]. In the first instance, the SNP dataset was subject to a filtering process using dartR to remove potentially erroneous SNPs. Monomorphic SNPs were excluded followed by the removal of SNPs with a reproducibility of <95%, a call rate of <90% (i.e., SNPs which have 10% missing genotypes or greater), and secondaries.

Pairwise $F_{ST}$, estimated as θ [64], was calculated between the five putative populations (Cry3Bb1, Cry34/35Ab1Ab1, Cry3B1_Cry34/35Ab1Ab1, adapted to crop rotation, and non-resistant), along with observed (Ho) and expected (He) heterozygosity. Departure from Hardy–Weinberg equilibrium (HWE) was tested for each population using the function *gl.report.hwe* as implemented in the R package dartR [63], which includes Bonferroni correction for multiple testing. Using the function *gl.basic.stats* in dartR, overall basic population genetics statistics per locus, such as the observed ($H_O$) heterozygosity, ($F_{IS}$) inbreeding co-efficient per locus, and $F_{ST}$ corrected for the number of individuals, was undertaken. To summarize genetic similarity among populations, *gl.tree.nj* in dartR was used.

The Bayesian model-based clustering algorithm implemented in the STRUCTURE v 2.3.4 [65] Evanno method was employed to determine the genetic structure of the WCR populations investigated. Genetic clusters (*K*-values) ranged between 1 and 6 (1 more population than the total number of populations for the complete data set), and a series of 10 replicate runs for each prior value of *K* were analyzed. The parameter set for each run consisted of a burn-in of 10,000 iterations followed by 100,000 Markov chain Monte Carlo iterations based on the admixture model of ancestry with the correlated allele frequency model and the default parameters in STRUCTURE. The most suitable value of *K* was calculated using the Δ*K* method as used in Structure Harvester web version 0.6.94 [66], where the highest Δ*K* value was indicative of the number of genetic clusters.

The marker-based kinship matrix (*K*) was calculated with the same genotypes using the VanRaden method [67] and then used to create a clustering heat map of the association mapping panel in the GAPIT [68].

2.4.2. Geometric Morphometrics

Each of fourteen previously established landmarks [48] for the WCR were digitized using the software program tpsDIG v.2.16 [69], for which x, y coordinates were generated to investigate hindwing shape. Statistical analyses were performed using MorphoJ version 1.06d [70]. Landmark coordinates were determined, and shape information was extracted using a full Procrustes fit [70]. Principal component analysis (PCA) was used to visualize hindwing shape variation in relation to the development of resistance [71]. PCA was based on the covariance matrix of individual hindwing shape. To visualize the average change in *Bt*-resistant strains, a covariance matrix of the average data (for all specimens, regardless of sex) was created. A PCA of the averaged data was used to better visualize shape morphology [72]. To compare morphological relationships between *Bt*-resistant and non-resistant populations, a canonical analysis of variance (CVA) was performed in order to calculate the morphological relationship between groups using the Mahalanobis and Procrustes distances. Mahalanobis and Procrustes morphological distances were calculated and reported with their respective *p*-values after a permutation test (10,000 runs). Finally, a multivariate regression of shape versus centroid size was performed to confirm whether size had an allometric effect [73].

**3. Results**

*3.1. Genetic Data*

3.1.1. Population Diversity Metrics

From the 29 WCR genotyped, 25,304 SNPs were detected. The 90% call rate filter then removed 13,852 SNPs from the data set. Following this, the minor allele fre-

quency filter, SNPs with frequencies <1%, hence removed another 3555 SNPs. Filtering for monomorphs, secondaries, and reproducibility set at 95% removed 772 SNPs. For final analyses, 7125 SNPs were used.

The overall population estimate was applied, and moderate observed heterozygosity ($H_O$) was observed across all loci, with an estimated value of $H_O = 0.325$. Moderate genetic diversity, estimated by expected heterozygosity ($H_E$), was observed with an estimated value of $H_E = 0.302$. Moderate inbreeding was observed ($F_{IS} = 0.121$). There were no significant deviations from HWE for all loci. The low overall value of the genetic structure ($F_{ST} = 0.0181$) estimated for the five populations suggested a lack of genetic differentiation amongst them as a whole.

Heterozygosity estimates ($H_O$ and $H_E$) over all loci and populations were very similar. The average $H_O$ per population ranged from 0.315 (non-resistant) to 0.338 (Cry3Bb1_Cry34/35Ab1), while average $H_E$ ranged from 0.315 (Cry34/35Ab1) to 0.349 (Cry3Bb1_Cry34/35Ab1) (Table 2). Moderate levels of genetic diversity across all populations were therefore suggested.

**Table 2.** Expected heterozygosity (He) and observed heterozygosity (Ho) values for western corn rootworm populations over all loci.

|  | No. of Individuals | No. of Loci | Ho | He |
|---|---|---|---|---|
| Cry3Bb1 | 7 | 6487 | 0.3203 | 0.3296 |
| Adapted to crop rotation | 4 | 6610 | 0.3352 | 0.3464 |
| Cry34/35Ab1 | 6 | 6247 | 0.3165 | 0.3158 |
| Cry3Bb1_Cry34/35Ab1 | 5 | 6562 | 0.3380 | 0.3494 |
| Non-resistant | 7 | 6261 | 0.3149 | 0.3170 |

Distribution of heterozygous WCR genotypes and SNP markers revealed moderate values of heterozygosity in 25 individuals out of 28, with heterozygosity <0.35 (Figure 2).

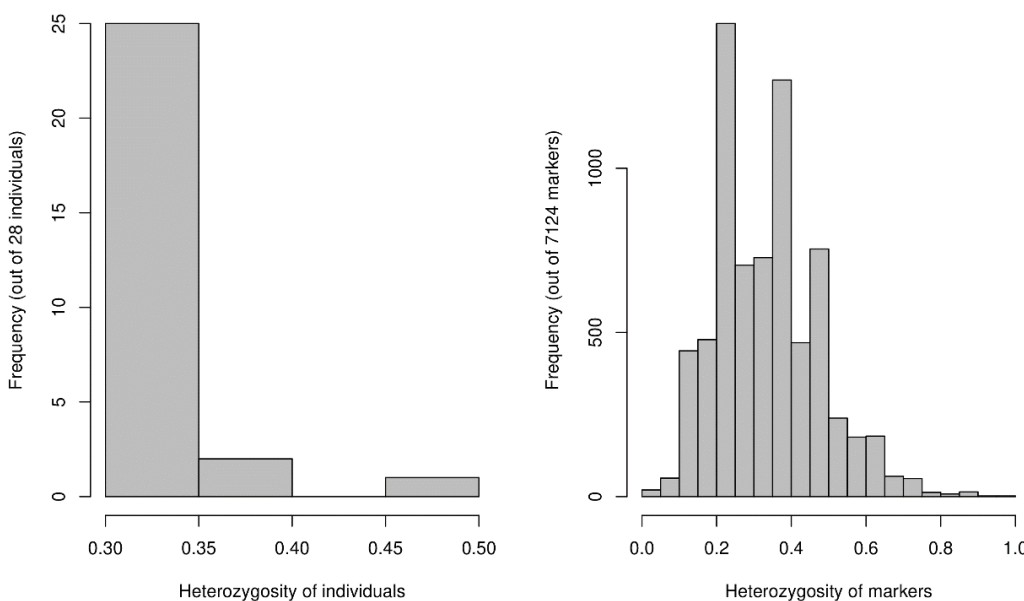

**Figure 2.** Frequency of heterozygous genotypes and heterozygosity of 7125 SNP markers.

In contrast, pairwise genetic structure does however show differentiation between pairwise population comparisons (Table 3). Pairwise $F_{ST}$ θ estimates ranged from 0.0021 (non-resistant population versus Cry3Bb1 resistant population) to 0.0531 (Cry34/35Ab1 resistant population versus Cry3Bb1_Cry34/35Ab1 resistant population). Cry34/35Ab1 and Cry3Bb1_Cry34/35Ab1 populations showed the greatest genetic differentiation with respect to all other populations.

**Table 3.** Population pairwise estimates of fixation index ($F_{ST}$).

|  | Cry3Bb1 | Adapted to Crop Rotation | Cry34/35Ab1 | Cry3Bb1_Cry34/35Ab1 |
|---|---|---|---|---|
| Cry3Bb1 |  |  |  |  |
| Adapted to crop rotation | 0.0028 |  |  |  |
| Cry34/35Ab1 | 0.0250 | 0.0242 |  |  |
| Cry3Bb1_Cry34/35Ab1 | 0.0238 | 0.0333 | 0.0531 |  |
| Non-resistant | 0.0021 | 0.0110 | 0.0206 | 0.0286 |

### 3.1.2. Genetic Structure

STRUCTURE analysis revealed $\Delta K = 3$ was the most likely number of clusters or populations present within the sampled US WCR individuals (Figure 3). Beetles were assigned to three clusters in consultation with results from STRUCTURE (Figure 4). Along with the results of the kinship analysis with the genetic clustering, a heat map of kinship matrix for evaluating the genetic differences among WCR genotypes was generated. Kinship coefficients between pairs of WCR genotypes varied very little on a scale of $-1$ to 1. However, the kinship matrix obtained from DArTseq SNP markers resulted in three distinct groups (Figure 5).

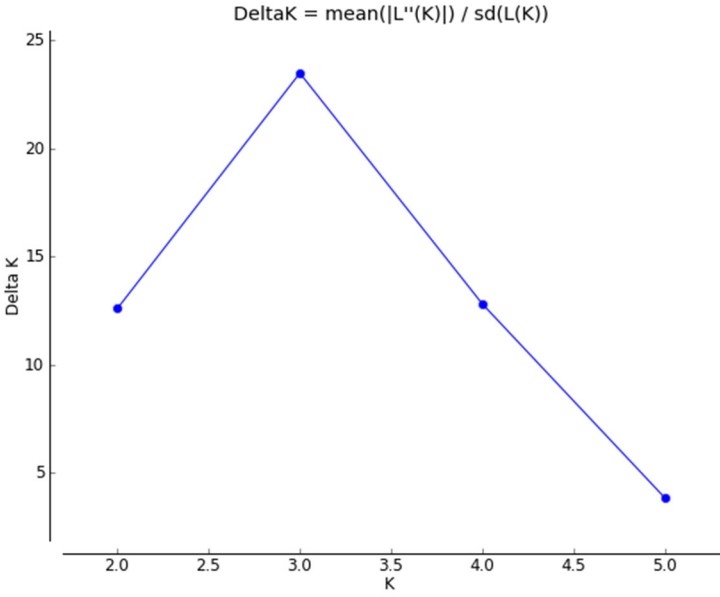

**Figure 3.** Results from Structure Harvester analysis to reveal the most likely value of *K* based on STRUCTURE results.

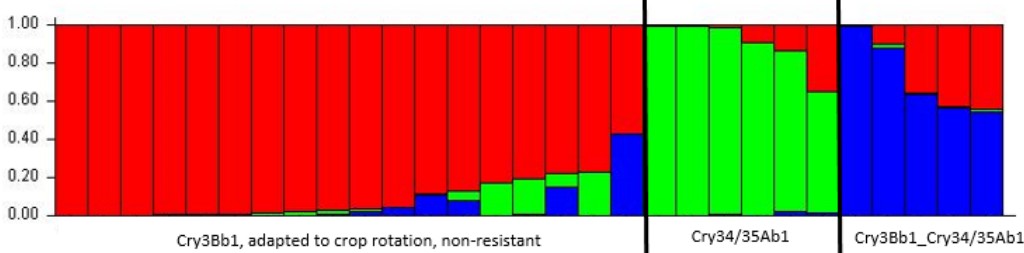

**Figure 4.** Determination of the optimal value of $K = 3$ and population structure of 29 WCR genotypes using DArTseq SNP markers.

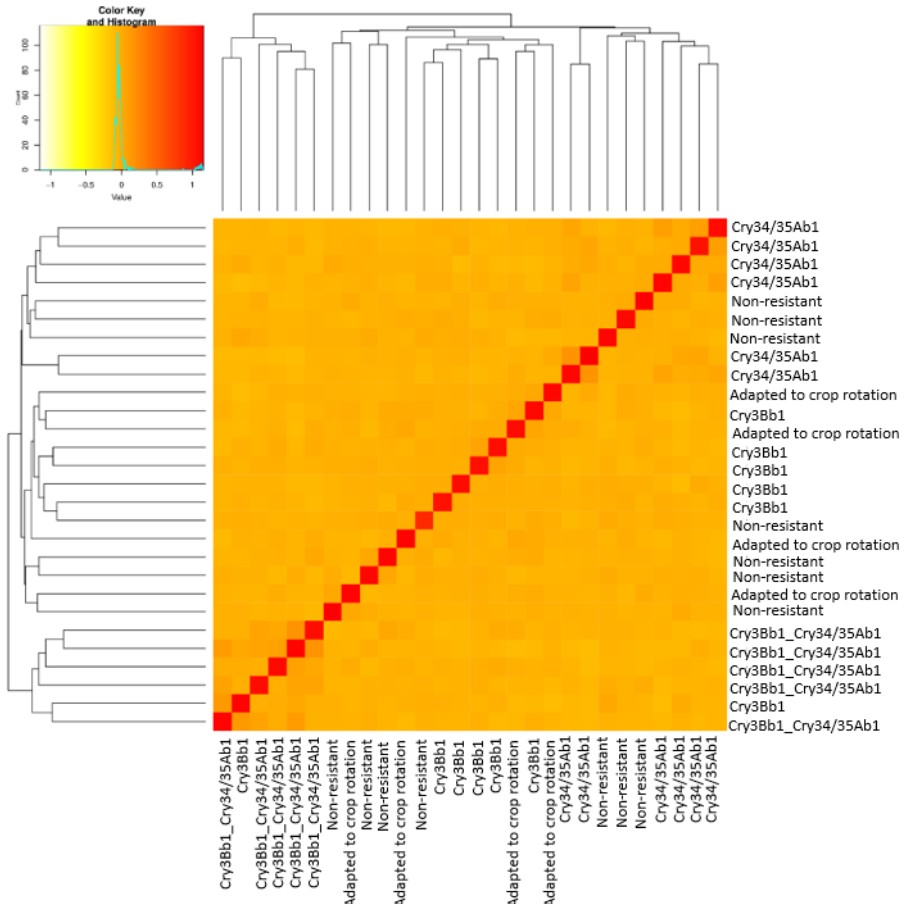

**Figure 5.** Heat map plot of kinship matrix using average linkage clustering based on SNP markers depicts the existence of three different groups among WCR genotype.

Further analysis of genetic structure using neighbor-joining (NJ) cluster analysis differentiated WCR genotypes into tree clusters (Figure 6). Cluster I was the largest, and it comprised 18 genotypes that included non-resistant individuals, Cry34/35 and Cry3Bb1 resistant. Cluster II contained individuals with combined *Bt* toxins Cry3Bb1 and Cry34/35 toxin, and Cluster III contained individuals adapted to crop rotation.

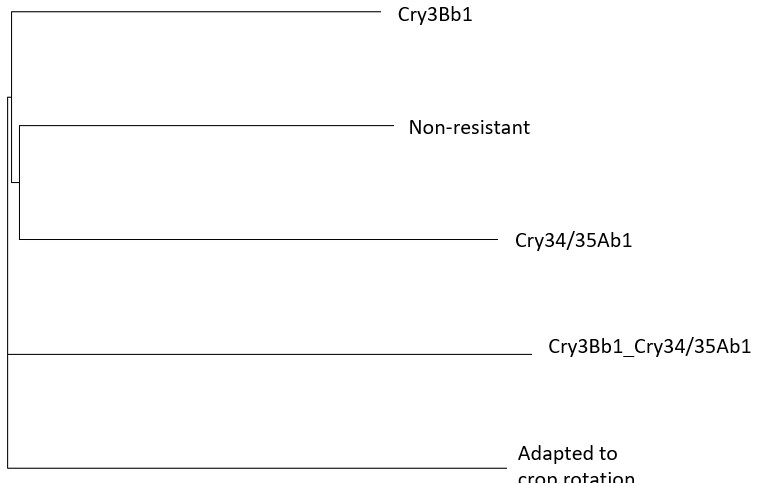

**Figure 6.** The neighbor-joining cluster analysis using DArTseq SNP markers for grouping 29 WCR genotypes.

### 3.2. Geometric Morphometrics

To avoid measurement error in our results, we calculated a Procrustes ANOVA showing that the mean square for individual variation exceeds the measurement error for wing shape (MS centroid size individuals: 0.000002 < 0.000107 MS centroid size error; and $7.0284 \times 10^6$ MS shape individuals <$7.428 \times 10^5$ MS shape error), so we can retain the following results. A multivariate regression analysis was performed before all the subsequent statistical analyses, discarding any allometric effect on the data (% predicted: 0.8033%).

The PCA of the hindwing shape showed an accumulation of the shape variation in a very few number of dimensions. The first three PCs accounted for 51.246% (PC1 = 21.12%; PC2 = 17.18%; PC3 = 12.93%) of the total shape variation and provided an approximation of the total amount of hindwing shape variation. After averaging the shape variation between the different populations, the population with Cry34/35Ab1 toxin was localized at the left of the PCA closer to the wing shape phenotype of the Cry3Bb1 but far away from the resistant and non-resistant populations where the latter was similar to the population of the combination Cry3Bb1_Cry34/35 (Figure 7).

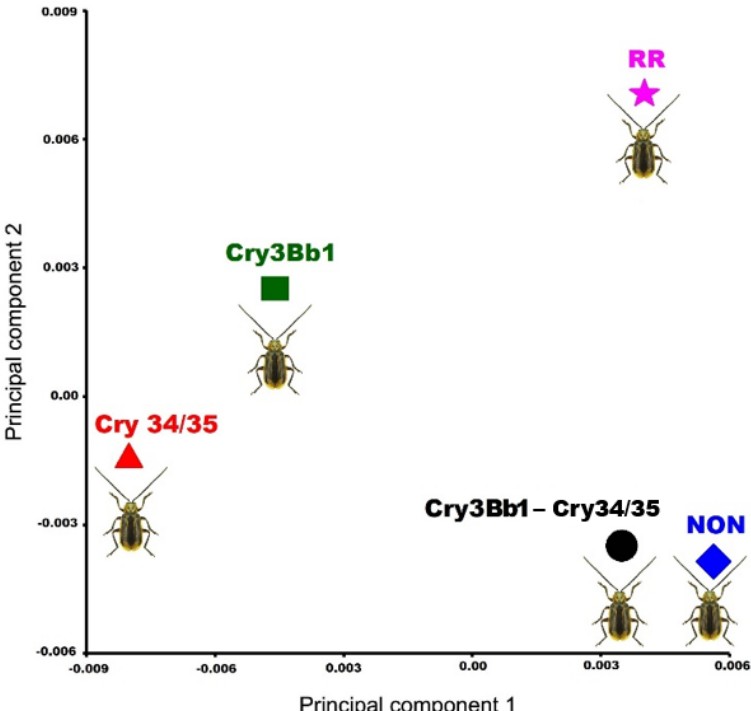

**Figure 7.** Principal component analysis of the hindwing average shape between different populations: resistant to the toxins, adapted to crop rotation, and non-resistant *Diabrotica virgifera virgifera.* Color and sign code: red triangle: Cry34/35Ab1 resistant population; green square: CryBb1 resistant population; pink star: population adapted to crop rotation (RR); black circle: CryBb1—Cry34/35Ab1 resistant population; and blue rhomboid (NON): non-resistant population.

Procrustes ANOVA showed clear significant differences between the hindwings size and shape between populations (Table 4).

In order to graphically visualize the differences, the CVA maximized the variance between groups, finding similar results with the genetic type in which the population of Cry34/35Ab1 separated from the non-resistant populations (Figure 8). Finally, significant differences (using the different morphometric distances) were found between populations after a permutation was run (Table 5).

**Table 4.** Procrustes ANOVA for both centroid size and wing shape of *Diabrotica virgifera virgifera*, Sums of squares (SS) and mean squares (MS) are in units of Procrustes distances (dimensionless).

| Centroid Size | | | | | | | |
|---|---|---|---|---|---|---|---|
| Effect | SS | MS | df | *F* | P (param.) | | |
| Toxins | 1,135,911.475839 | 283,977.869 | 4 | 21.6 | <0.0001 | | |
| Individual | 3,431,958.659351 | 13,149.26689 | 261 | 45.74 | <0.0001 | | |
| Residual | 56,921.18152 | 287.480715 | 198 | | | | |
| Shape | | | | | | | |
| Effect | SS | MS | df | F | P (param.) | Pillai tr. | P (param.) |
| Toxins | 0.03076466 | 0.0003204652 | 96 | 4.7 | <0.0001 | 1.12 | <0.0001 |
| Individual | 0.42691601 | $6.81539 \times 10^5$ | 6264 | 2.36 | <0.0001 | 17.64 | <0.0001 |
| Residual | 0.13725163 | $2.88829 \times 10^5$ | 4752 | | | | |

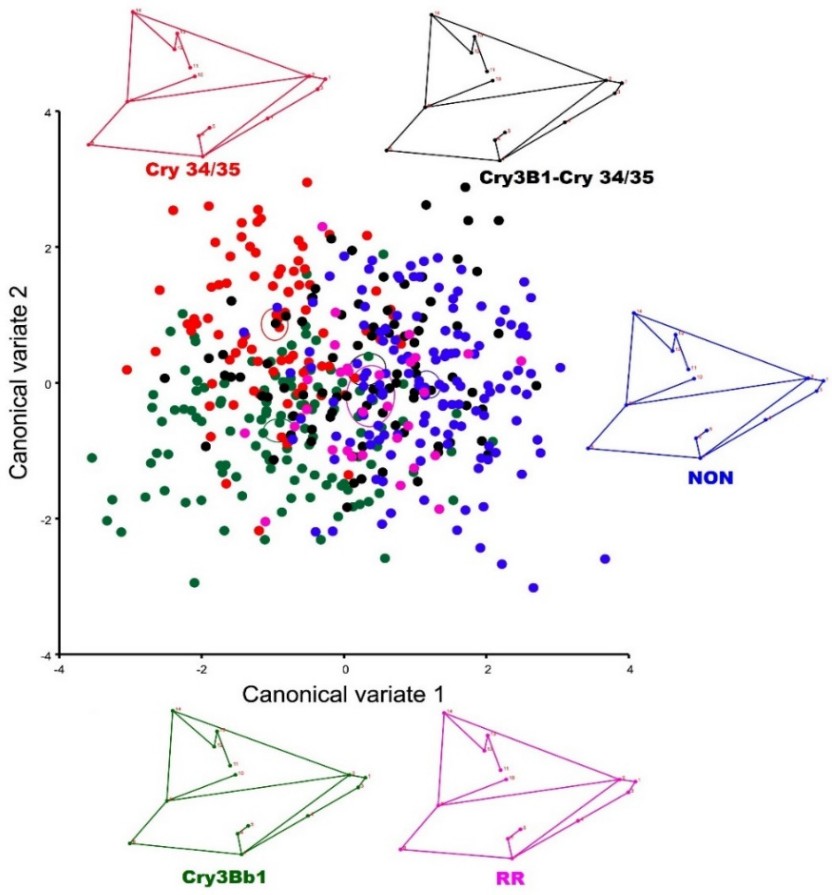

**Figure 8.** Canonical variate analysis of the hindwing shape between different populations resistant to the toxins: adapted to crop rotation and non-resistant population in *Diabrotica virgifera virgifera*. Color and sign code: red Cry34/35Ab1 resistant population; green CryBb1 resistant population; pink population adapted to crop rotation (RR); black CryBb1-Cry34/35Ab1 resistant population; and blue (NON): non-resistant population.

**Table 5.** Mahalanobis and Procrustes distances between groups obtained from canonical variate analysis. *: $p < 0.05$; **: $p < 0.001$.

| | Mahalanobis Distances | | |
|---|---|---|---|
| | Cry34/35 | Cry3Bb1 | NON |
| Cry3B1_Cry34/35 | 1.8022 ** | | |
| Cry3Bb1 | 1.5633 ** | 1.7142 ** | |
| NON | 2.3832 ** | 1.3276 ** | 2.2068 ** |
| RR | 2.305 ** | 1.6339 ** | 1.9881 ** |
| | Procrustes Distances | | |
| | Cry34/35 | Cry3Bb1 | NON |
| Cry3B1_Cry34/35 | 0.0135 ** | | |
| Cry3Bb1 | 0.0107 ** | 0.0124 ** | |
| NON | 0.0155 ** | 0.0069 * | 0.013 ** |
| RR | 0.0154 ** | 0.0118 * | 0.0132 ** |

## 4. Discussion

In this research we aimed to find a reliable pattern of differences related to resistance type using genetic and geometric morphometric analyses. For population structure analysis, we used DArTseq SNP markers. One of the questions we were interested in was whether resistant WCR populations differ at the genetic level. We found no significant evidence of high genetic diversity in any of the assumed populations. However, the estimated values were congruent with moderate genetic diversity across the genotyped beetles. The STRUCTURE revealed three genetic clusters. This classification was also supported by the VanRaden kinship algorithm, where Cry3Bb1_Cry34/35Ab1 individuals and Cry34/35Ab1 were separated from Cry3Bb1 adapted to crop rotation and non-resistant individuals, although some non-resistant individuals mixed between Cry34/35Ab1, which could be due to the normal evolutionary process. The fact that Cry3Bb1 non-resistant and adapted to crop rotation populations are mixed suggests that they are genetically similar (Figure 4). The neighbor-joining tree separated the individuals adapted to crop rotation, which is to be expected given that the first evolved resistance (not including insecticides) was to crop rotation [12]. Afterwards, all other resistance evolved, and we can see that clearly in this result. The fact that the non-resistant population is not separated could be due to an evolutionary process, as we mentioned earlier.

High-throughput sequencing has provided deeper insight into the molecular mechanisms of resistance [74]. It allowed us to find that many point mutations are found in different genes, suggesting that these mechanisms can occur simultaneously, making it more difficult to understand which one is really responsible for the resistance phenotype [75,76]. In our research, we focused on resistant populations, and we determined that there was some variability between them, but there was no exact pattern. Recent molecular studies show us that different sets of genes are involved in resistance [76–79], which makes it unlikely that universal markers of resistance can be developed to accurately determine the likelihood of a population becoming resistant to a particular compound [75,77,79]. A different number of genes may be involved in resistance, and individuals within a population exhibit different evolutionary patterns of resistance evolution. Therefore, resistance can be found throughout the whole genome, but it is not conditioned by the differences. However, certain shifts could be a warning that some changes in the genome have occurred. Through estimates of genetic diversity, population structuring, and genetic relatedness between individuals, information on the effectiveness of control strategies can be obtained, and recommendations to improve the efficacy of control programs may be possible.

The actual sample size of each site does not need to be large when using SNPs. SNP markers provide the power, not the sample size, as SNPs have genome-wide coverage and there end up being many thousands of SNPs by the time genotyping is complete [80]. The paper by Trask et al. [81] states, "Given that each SNP marker has an individual

evolutionary history, we calculated that the most complete and unbiased representation of genetic diversity present in the individual can be achieved by including at least 10 individuals in the discovery sample set to ensure the discovery of both common and rare polymorphisms." The second paper by Li et al. [82], who also worked with beetles from the order Coleoptera, found that "a minimum sample size of 3–8 individuals is sufficient to dissect the population architecture of the harlequin ladybird, *Harmonia axyridis*, a biological control agent and invasive alien species." They also estimated the optimal sample size for accurately estimating genetic diversity within and between populations of *Harmonia axyridis*. They determined that six individuals are the minimum sample size required.

Wing morphology (size and shape) is the most important trait of an insect's dispersal capacity. For this reason, the integration of different techniques to understand the plasticity and variation of this trait is vital to understanding how they adapt to new environments and to coordinating strategic planning ahead of possible new invasions [3]. Different types of wing morphotypes have been studied to determine the dispersal capabilities of flying insects [83–85]. Le et al. [86] found that narrowed wings in beetles are more efficient for flapping low-level flights. Additionally, for *D. v. virgifera*, wing shape has been identified as a very good trait to measure in different agronomic studies, including studies of life history (sexual dimorphism) and interspecific and intraspecific shape variation [47–49], and wing shape has also been a useful variable when combined with other monitoring tools (genetics (e.g., microsatellites) and traditional traps (e.g., pheromones)) [3].

Mikac et al. [46] showed that beetles adapted to crop rotation had broader wings (cf. susceptible beetle). Mikac et al. [45] expanded the use of differences in hindwing size and shape to examine changes in WCR associated with the development of resistance, specifically to examine potential differences between (*Bt*)-resistant, non-resistant (or susceptible), and adapted to crop rotation populations in the US. In general, the hindwings of non-resistant beetles were significantly more elongated in shape and narrower in width (chord length) compared with beetles resistant to *Bt* maize or crop rotation. This result was confirmed by our study. Mikac et al. (2019) did not separate the *Bt*-resistant populations in their study, but considered them as one population. Therefore, in our study, we separated all *Bt*-resistant populations to see the differences between them. Cry3Bb1_Cry34/35Ab1 individuals had the broader shape and a more robust wing with an expansion of landmark 14 and a contraction of landmark 9. Cry3Bb1 individuals had the narrower wings, while individuals resistant to Cry34/35Ab1 had similar but smaller wings, distinguished by the expansion of landmarks 3 and 4. The more stable and elongated wing shape was that of the population adapted to crop rotation, in which there was an extension to landmarks 1 and 2 to the left and an elongation to landmark 9 to the right. The non-resistant population is also slightly wider than the population of Cry3Bb1-Cry34/35Ab1, with the movement of landmarks 14 and 2 also slightly to the right and the wider shape that is also produced by the movement of landmark 7 to the upper left. Elongated wings are more aerodynamic and are considered to be involved in migratory movement [46]. Mikac et al. [46] also suggested that this could be a useful invasive dispersal strategy for mated females. In our research, individuals adapted to crop rotation had more stable and elongated wings, suggesting that these individuals could fly long distances. Such differences may impact upon the dispersal or long-distance movement of resistant and non-resistant beetles. Understanding which beetle morphotype is the superior flyer and spreader has implications for managing WCR through integrated resistance strategies. These findings confirmed GM as a reliable technique for resistance detection. In this study, we aimed to confirm the results from SNPs markers with GM. We found that geometric morphometric tools could provide important clues to differentiate resistant and non-resistant populations. One of the principal results was the similarity of the hindwing shape variation between the population after the STRUCTURE analysis, where using both monitoring techniques showed that the more differentiated population was the resistant Cry34/35Ab1.

Here we describe a possibility that combining genetic and geometric morphometrics could be a reliable technique that can be used to reveal differences among WCR populations.

Hence, geometric morphometrics can be used as a biomarker for resistance detection as part of a larger integrated resistance management strategy for western corn rootworm.

In Croatia, WCR have been investigated in detail (traditional monitoring, genetic monitoring, and GM monitoring), and knowledge about dispersal and adaptive abilities of these invasive insects is well known [3,47,87,88]. Our future work will focus on populations collected in intensive maize-growing areas in Croatia, where WCR populations have become established since their introduction 30 years ago. We will use the comparative techniques presented in this paper to determine whether Croatian populations are potentially resistant and which US WCR population was the source population for Croatia and Europe. This knowledge would help to detect resistant individuals that might invade geographical areas where they are not yet present (e.g., beetles adapted to crop rotation invading Europe where such variants are not present). Such information is very important for biosecurity measures, resistance management, and future control strategies for this pest worldwide.

**Author Contributions:** Conceptualization, M.K.B. and D.L.; data curation, M.K.B., K.M.M., H.A.B. and R.B.; formal analysis, M.K.B., H.A.B. and D.L.; funding acquisition, R.B.; investigation, M.K.B., K.M.M., H.A.B., R.B. and D.L.; methodology, M.K.B., K.M.M., H.A.B., R.B. and D.L.; project administration, R.B.; resources, R.B.; software, M.K.B. and H.A.B.; supervision, K.M.M., R.B. and D.L.; validation, R.B. and D.L.; visualization, M.K.B. and H.A.B.; writing—original draft, M.K.B. and H.A.B.; writing—review and editing, K.M.M., R.B. and D.L. All authors have read and agreed to the published version of the manuscript.

**Funding:** This study was supported by the Croatian Science Foundation through the project Monitoring of Insect Pest Resistance: Novel Approach for Detection, and Effective Resistance Management Strategies (MONPERES) (IP-2016-06-7458) and the young researchers' career development project training of new doctoral students (DOK-01-2018).

**Institutional Review Board Statement:** Not applicable. Western corn rootworm is an established pest of maize in the USA and Southern Europe. No special permission was needed for its collection in this study.

**Acknowledgments:** The authors thank Wade French and Chad Nielson for Bt-maize-resistant rootworm colonies from South Dakota and Joseph Spencer for providing field-collected beetles adapted to crop rotation from Illinois. The authors are very grateful to Reza Talebi and João Paulo Gomes Viana for their help and advice with data analysis. The authors also thank colleague Zrinka Drmić and student Patricija Majcenić for help in preparing the WCR wings.

**Conflicts of Interest:** The authors declare no conflict of interest.

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
