# Peer review of "Genetic and Morphological Approach for Western Corn Rootworm Resistance Management"

_agriculture, doi:10.3390/agriculture11070585_

Round 1

Reviewer 1 Report

The manuscript titled “Genetic and morphological approach for western corn rootworm resistance management” aims to develop a novel approach for managing western corn rootworm. The authors found no significant difference in whole genome among Bt-resistant and non-resistant populations, whereas some differences in hindwings of these populations were found. In general, there are some errors within the text that must be addressed and some clarifications that should be made. 

Major comments

  1. One of my major concerns is that no analysis was used to determine relationship between genetic and morphological results in this study?
  2. The information on western corn rootworm populations is missing? 05 different WCR populations were analyzed, including 03 WCR resistant populations to Bt toxins (Cry3Bb1, Cry34/35Ab1 (?), Cry3Bb1 and Cry 34/35Ab1), and 01 rotation resistant population, and 01 susceptible population to (all) Bt toxins? Please provide information on these WCR colonies? For example, where they were collected? How laboratory populations were selected and developed? How long or many generations they are reared and maintained in laboratory? What are resistant or susceptible capacities of these colonies (LC50) to corresponding Bt proteins? Or providing citations if applicable for that important information.
  3. Please also explain why you compare a field collected WCR population (rotation resistant population) with 4 laboratory maintained WCR populations. The two different conditions (field, lab-rearing) possibly contribute to differences in wing shapes and sizes since beetles are reared in small cages in the lab compared with natural environment in the field.
  4. Why no distinctions between male and female beetles were made in geometric morphometric analysis? Since females with eggs can carry heavier payloads compared to males, females and males may have hind wings with different sizes or shapes.
  5. The key finding of this paper could be wing shapes and sizes of beetles in selected and unselected Bt toxins, please provide figure(s) describing shapes/sizes of hindwings and the differences among different WCR populations.
  6. Information on Materials and Methods section is not adequately, and thereby this study cannot be replicated based on the materials and methods provided. Please provide information on experimental design (e.g., how to design treatments, how many biological replicates for each experiment). Please describe which data were used for geometric morphometrics and how they were measured and collected?

Please find the pdf other comments.

Author Response

Dear reviewer,

I would like to thank you for your time and effort in reading our article. Also, for your comments and suggestions.

Response to Reviewer 1 comments

  1. One of my major concerns is that no analysis was used to determine relationship between genetic and morphological results in this study?

Response: The methods presented in this study (genetic and geometric morphometrics) have been used with the main aim to identify resistance pattern e.g. changes caused by resistance development in specific insect population. As is already known that metric properties are under the influence of both environmental and genetic factors. Environment typically acts primarily on size (Glasgow 1961), and then on shape. Morphometric marker shows current and recent genetic changes (Bouyer at al. 2007).

Nowdays, morphometric analyses are in increasingly widespread use in conjunction with analyses of molecular data. Examples of such combined uses of data include mapping of shape data onto phylogenies inferred from DNA sequences, quantitative genetic analyses based on genealogies extracted from molecular marker information, or assessments of the effects of hybridization on shape asymmetry.

The most widespread approach in geometric morphometrics is to represent each specimen by the relative positions of morphological landmarks that can be located precisely and establish a one-to-one correspondence among all specimens included in the analysis, in our study on WCR hind wings. Shape is then defined as all the geometric information about a configuration of landmarks except for its size, position and orientation (Dryden & Mardia 1998). The shape information is extracted by a procedure called Procrustes superimposition, which removes variation in size, position and orientation from the data on landmark coordinates, and which is at the core of geometric morphometrics (Goodall 1991; Bookstein 1996; Dryden & Mardia 1998; Zelditch et al. 2004). The coordinates of the superimposed landmarks can be used in multivariate statistical analyses to address a wide range of biological questions (e.g. Klingenberg 2010).

The current MORPHOJ software includes methods for finding shape variables of maximal or minimal heritability, useful for identifying genetic constraints on evolution, and for simulating hypothetical selection scenarios that can help in assessing genetic integration in a structure (Klingenberg & Leamy 2001; Martı´nez-Abadı´as et al. 2009; Klingenberg et al. 2010).

Quantitative genetic studies are increasingly feasible, even in natural populations when pedigree information is available from behavioural observations or molecular markers (McGuigan 2006; Wilson et al. 2010), and the methods for such studies have been extended to shape analyses (Klingenberg & Leamy 2001; Klingenberg & Monteiro 2005; Myers et al. 2006; Klingenberg et al. 2010a).

In our research no specific resistance marker is known, so both methods have been used to work together to estimate variability/changes in SNPs and wing shape that could address resistance development. 

  1. The information on western corn rootworm populations is missing? 05 different WCR populations were analyzed, including 03 WCR resistant populations to Bt toxins (Cry3Bb1, Cry34/35Ab1 (?), Cry3Bb1 and Cry 34/35Ab1), and 01 rotation resistant population, and 01 susceptible population to (all) Bt toxins? Please provide information on these WCR colonies? For example, where they were collected? How laboratory populations were selected and developed? How long or many generations they are reared and maintained in laboratory? What are resistant or susceptible capacities of these colonies (LC50) to corresponding Bt proteins? Or providing citations if applicable for that important information.

Response: added to the manuscript L142- 153

  1. Please also explain why you compare a field collected WCR population (rotation resistant population) with 4 laboratory maintained WCR populations. The two different conditions (field, lab-rearing) possibly contribute to differences in wing shapes and sizes since beetles are reared in small cages in the lab compared with natural environment in the field.

Response:

Study from Mikac et al. 2019 analyzed hindwings from rotation resistant, Bt resistant and susceptible beetles. Difference in wing shape as related to rearing method (laboratory versus field collected rotation-resistant samples) was explored as it can affect wing morphology principally when the properties of the rearing system are related directly to the developmental stability of the organism (Gerard et al. 2018). However, the minimal differences found between rotation-resistant laboratory versus field-collected populations suggests that rearing system was not the main driver of the small differences observed. That’s why we excluded the possibility that different conditions (field, lab-rearing) possibly contribute to differences in wing shapes and sizes.

Furthermore, Kim et al. 2007 in their research used microsatellites to study genetic diversity in WCR laboratory colonies. They assess changes in genetic diversity in laboratory colonies of WCR maintained at NCARL. Their results suggest that diapause colonies maintained at NCARL are genetically similar to wild populations

  1. Why no distinctions between male and female beetles were made in geometric morphometric analysis? Since females with eggs can carry heavier payloads compared to males, females and males may have hind wings with different sizes or shapes.

Response: Sexual shape dimorphism was not an aim to evaluate in this article, there are some evidence about shape differences in female carrying eggs, but for this article we only focused in the general relationship.

  1. The key finding of this paper could be wing shapes and sizes of beetles in selected and unselected Bt toxins, please provide figure(s) describing shapes/sizes of hindwings and the differences among different WCR populations.

Response: We thanks the reviewer comments, the average shape of the hindwings were provided in the figure 8 using different colours.

  1. Information on Materials and Methods section is not adequately, and thereby this study cannot be replicated based on the materials and methods provided. Please provide information on experimental design (e.g., how to design treatments, how many biological replicates for each experiment).

Response: information added as suggested

  1. Please describe which data were used for geometric morphometrics and how they were measured and collected?

Response: the description was added L206-208; also new Figure 1 was added were is representation of the 14 morphological landmarks identified on the hind wings of WCR.

ABSTRACT

  1. Please explain how the difference in geometric morphometric results confirmed the genetic results?

Response: Geometric morphometrics tools can be used when morphological results are difficult to discern by normal observation. In this study, the results of genetics using a large portion of the genome confirmed distinct resistance and non-resistance populations in D.v. virgifera. Nevertheless, the genes are completely different from the morphological traits and the analyzes are independent. Since the wings are traits that this species uses to move between invasion sites, clear differences were found in their average shape from each population in terms of geometric variation, similar to genetically structured populations, two independent monitoring techniques therefore confirm the same results.

  1. Please explain how geometric morphometrics captured genetic patterns and how to apply geometric morphometrics to determine an estimation of population

Response: Metric properties are under the influence of both environmental and genetic factors.  Changes in an organism's genotype takes much longer to manifest than on its phenotype. Morphometric marker shows current and recent genetic changes (Bouyer at al. 2007). If we detect changes on wings we can assume there are some changes in population structure.

  1. Please explain how SNPs and geometric morphometrics were or can be combined?

Response: added in abstract

  1. Please provide information on the insights to pest management?

Response: added in abstract

RESULTS

  1. The genetic differences among WCR genotypes was generated. Describe the results, and what is the difference?

Response: added and described as suggested

  1. Both PCA accounted for 10.25% of data variation, indicating that models are not reliable and should not be used.

Response: PCA figure has been removed from the manuscript as suggested

  1. Explain why the cluster analysis does not agree with PCA analysis?

Response: PCA figure has been removed from the manuscript

  1. This information should be described in data processing and analysis (After averaging the shape variation…) Line 287

Response: This information was incorporated in the line 230-231 indicating that a PCA of the average data was used

  1. Figure 7. Why the same image of WCR beetle was used?

Response: Same image was used to have a nice representation of the graph, the analysis per se was made in the hind-wing

DISCUSSION

  1. The beetles from the laboratory were mixed by accident. Please explain. The susceptible data to Bt toxins of non-resistant population are needed for the explanation.

Response: The fact that the non-resistant population is not separated could be due to an evolutionary process. Beetles from the non-diapausing, susceptible colony, were provided by the USDA-ARS laboratory in Brookings, SD. This colony is derived from a field population collected prior to the release of transgenic crops, so this colony should be susceptible to all Bt proteins active against WCR (Ludwick et al., 2018).

  1. In our research rotation resistant individuals had more stable and elongated wings suggesting these individuals could fly to long distances. Data on this finding need to be included in the results.

Response: Data of this is provided in the average shape of Figure 7, variation of the shape is slightly different between groups for the reason a CVA was made, an extra figure for this analysis we think is not necessary itself.

  1. This knowledge would also help detect resistant individuals that might invade geographic areas where they are not yet present (i.e. rotation- resistant beetles invading Europe where such variants are not present). Please describe how to do that?

Response: Here we were looking to find a reliable pattern of difference related to resistance type. We will use compared techniques presented in this paper to determine if the Croatian populations are potentially resistant and which US WCR population was the source population for Croatia and Europe.

Added in the discussion part

Reviewer 2 Report

Manuscript Number ID: agriculture-1239626 

This study provides important insights about the emergence of western corn rootworm variants able to adapt to rotation and/or Bt maize using genetic and geometric morphometrics approaches. I think that the idea is really good, and the general approach to research also strong. However, I think there are a couple of important points should be addressed more carefully in this study. First of all, the authors should clarify in the Material and Methods why they used so few individuals for the genetic analyses (number of individuals for each representative population), they should discuss the possible gaps and how this choice possibly affected the results. Lastly, the discussion of geometric morphometric results is too speculative and not properly addressed. I strongly suggest to improved these 2 parts. There are several other suggestions I reported in the highlighted comment areas along the text in the pdf attached below, that I suggest to take into account in a revised version of the manuscript.

Author Response

Dear reviewer,

I would like to thank you for your time and effort in reading our article. Also, for your comments and suggestions.

Response to Reviewer 2 comments

This study provides important insights about the emergence of western corn rootworm variants able to adapt to rotation and/or Bt maize using genetic and geometric morphometrics approaches. I think that the idea is really good, and the general approach to research also strong. However, I think there are a couple of important points should be addressed more carefully in this study. First of all, the authors should clarify in the Material and Methods why they used so few individuals for the genetic analyses (number of individuals for each representative population), they should discuss the possible gaps and how this choice possibly affected the results. Lastly, the discussion of geometric morphometric results is too speculative and not properly addressed. I strongly suggest to improve these 2 parts. There are several other suggestions I reported in the highlighted comment areas along the text in the pdf attached below, that I suggest to take into account in a revised version of the manuscript.

ABSTRACT

  1. Genetic results showed that there were some differences among the resistant US populations. please specify which differences?

Response: added in abstract as suggested

INTRODUCTION

  1. thousands separator??

Response: changed

  1. please include it is also exotic for Europe

Response: included

  1. It is unusual to state that an insect developed resistance to cultural control method. I suggested to state that WCR adapted to crop rotation and developed resistance to Bt toxin ...

Response: changed as suggested

  1. I suggest "to the early detection of resistance or adaptation

Response: changed as suggested

  1. L84-85. This sentence is a little bit confusing, please clarify

Response: sentence is changed

  1. A verb missing in this sentence, please double-check and edit.

Response: checked and edited

  1. Please replace the highlighted text with: “genetic and geometric morphometric techniques on..."

Response: changed as suggested

  1. and predict?

Response: added as suggested

MATERIALS AND METHODS

  1. The author did not introduce these toxins in Introduction. Please describe here

Response: added to the manuscript L142- 153

  1. Please spell out the acronyms GM and SNP in the caption of the Table.

Response: added as suggested

  1. I have a concern on the minimum sample size for population genetic study using SNPS, the authors used apparently max 7 individuals in each population. Please justify this choice and discuss the possible limitations

Response:

The actual sample size of each location don’t need to be big when using SNPs because it’s the SNP markers that will give the power, rather than the sample size, because SNPs have a genome wide coverage and it will end up with many thousands of SNPs once the genotyping is complete.  Paper by Satkoski Trask et al. (2011) state ‘Given that each SNP marker has an individual evolutionary history, we calculated that the most complete and unbiased representation of the genetic diversity present in the individual can be obtained by incorporating at least 10 individuals into the discovery sample set, to ensure the discovery of both common and rare polymorphisms.’ The second paper is from Li et al. (2020), the worked also with beetle from Coleoptera order and they stated that ‘a minimal sample size of 3–8 individuals is sufficient to dissect the population architecture of the harlequin lady beetle, Harmonia axyridis, a biological control agent and an invasive alien species’. Also they estimated the optimal sample size for the accurate estimation of intra- and inter-population genetic diversity for Harmonia axyridis. They determined that six individuals are the minimum sample size required.

  1. By the Table 1, I assumed there are 5 populations? please specify here or above

Response: added and specified as suggested

RESULTS

  1. The table caption should be self-explicative. Please spell out the acronyms and put He and Ho in parenthesis.

Response: added as suggested

  1. L235-236. I suggest to report the results clearly: it would be 25 individuals out of 28 with heterozygosity <0.35 ...

Response: change as suggested

  1. This belong to material and methods. Please move it there

Response: changed as suggested

  1. The authors should report here the variance explained by each axis.

Response: the figure has been deleted on suggestion of other reviewer

  1. This is should be figure 5 as it is cited before. Please amend it.

Response: changed as suggested

  1. This decodification is not useful at all. Please describe the acronym properly here. For example: "blue rhomboid (NON): non-resistant population"

Response: Decodification is improved as suggested.

DISCUSSION

  1. Authors should discuss if the low number of individuals considered here for each population could possibly affects these results

Response: paragraph about the low number of individuals has been added in discussion

  1. As pointed out above, I think it is wrong to say that population are resistant to the rotation, rather they are adapted to the rotation. Please fix this concept along the entire paper.

Response: changed in the whole paper as suggested

  1. Please improve by including examples of genetics tools and replace "traditional types" with "traditional traps (e.g. pheromones)"

Response: added and changed as suggested

  1. replace with "individual adapted to rotation"

Response: changed as suggested

  1. L387-389. It would be more interesting to increase the study in WCR. I think that this sentence is not useful for the purpose of this paper in the Discussion section. I suggest to delete it.

Response: sentence deleted as suggested

  1. The word demonstrated in not correct here. Actually the authors reported in lines 381-383 that they suggest a correlation between elongated wings and migration capability, as Mikac et al. for this reason this sentence is too speculative. The authors should use caution with the use of the word "demonstrated". Please change the sentence accordingly.

Response: improved as suggested

  1. L400-401. There is possibility to couple direct measurements of dispersal capability along with GM approach? Please clarify and add here some comments about that.

Response: clarified as suggested

Round 2

Reviewer 1 Report

The authors have addressed most of my minor concerns, and this manuscript is much improved. However, they failed to convince me that they know the starting materials used in this manuscript.

The authors cite Zukoff et al. (2016) as the source of the resistant insects (Line 147), but I know with 100% certainty that none of these insects ever went to NCARL. Ludwick et al. (2018) also has nothing to do with a susceptible strain (Line 149). Tables 1 and 3 discuss resistance to Cry3Bb1, Cry34/35Ab1, and apparent resistance to both (Lines 162, 273). However, the laboratory at NCARL has not published anything on resistance to anything other than Cry3Bb1. Oswald et al. 2011 did publish on Cry3Bb1 resistance developed at NCARL, but this paper is not even cited by the authors. How can the authors claim to have not only resistance to Cry34/35Ab1, but also resistance to SmartStax (Cry3Bb1+Cry34/35Ab1) when NCARL has never published on the latter two?

Moreover, how can the authors publish on wing shape when all of these populations have been in the lab for many generations? Under laboratory rearing conditions, adult western corn rootworms are reared in small cages, which likely contributes to differences in wing shapes. The differences in wing shapes that the authors found might or might not be driven by the effect of Bt toxins?

Given these concerns, I regrettably must suggest that the authors resubmit the manuscript when they have acquired information on the insect materials, data that support their findings, and make the suggested editorial change.

Author Response

REVIEWER 1_2nd round

The authors have addressed most of my minor concerns, and this manuscript is much improved. However, they failed to convince me that they know the starting materials used in this manuscript.

The authors cite Zukoff et al. (2016) as the source of the resistant insects (Line 147), but I know with 100% certainty that none of these insects ever went to NCARL. Ludwick et al. (2018) also has nothing to do with a susceptible strain (Line 149). Tables 1 and 3 discuss resistance to Cry3Bb1, Cry34/35Ab1, and apparent resistance to both (Lines 162, 273). However, the laboratory at NCARL has not published anything on resistance to anything other than Cry3Bb1. Oswald et al. 2011 did publish on Cry3Bb1 resistance developed at NCARL, but this paper is not even cited by the authors. How can the authors claim to have not only resistance to Cry34/35Ab1, but also resistance to SmartStax (Cry3Bb1+Cry34/35Ab1) when NCARL has never published on the latter two?

Moreover, how can the authors publish on wing shape when all of these populations have been in the lab for many generations? Under laboratory rearing conditions, adult western corn rootworms are reared in small cages, which likely contributes to differences in wing shapes. The differences in wing shapes that the authors found might or might not be driven by the effect of Bt toxins?

Dear reviewer,

thank you very much for your valuable comments and your time. Without your help we probably would not have realized the mistake we made.

In personal communication with Chad Nielson (USDA-ARS-NCARL), we realize there was a misunderstanding regarding the WCR colonies. The non-resistant colony is from the NCARL laboratory. However, the Bt-resistant colonies are collected in the field on Bt maize expressing different toxins. We have changed this in our manuscript (L138 – 141) and will be more careful in the future.

The non-resistant laboratory population was collected near the town of Trent South Dakota in Moody county the year of 1987.  It has been in continuous rearing since that time with no mixing from any other collections.  It is about one generation per year.  However, they maintain the colony on a continuous basis and have all growth stages continuously year round.  A single beetle would complete one generation a year if tracked because eggs spend several months in cold storage.  The original beetles were selected in corn fields or on the margin of corn fields and the adult beetles returned to the lab.  Eggs were collected and placed in cold storage to simulate winter.  The non-resistant colony is reared in soil on corn roots and the adult beetles receive an artificial diet.  They try to keep the rearing protocol “field like” to keep it “wild”. 

Regarding your question about wing shape, I think there is no problem now considering that the WCR individuals are not from the lab (except for the non-resistant population). However, we would like to reiterate that in the Mikac et al. 2019 study, hindwings of rotation-resistant, Bt-resistant, and susceptible beetles were analyzed. Differences in wing shape as related to rearing method (laboratory versus field-collected rotation-resistant samples) were examined because they may affect wing morphology, especially when the characteristics of the rearing system are directly related to the developmental stability of the organism (Gerard et al. 2018). However, the minimal differences found between rotation-resistant laboratory and field-collected populations suggest that the rearing system was not the main driver of the observed differences. Therefore, we excluded the possibility that different conditions (field, laboratory rearing) may contribute to differences in wing shapes and sizes.

Reviewer 2 Report

The authors replied point by point to the criticism raised in the first revision. I believe that the current version was deeply improved.

Author Response

Dear reviewer,

We would like to thank you once more for your time and effort in reading our article.